# Glass Transition Behaviors of Poly (Vinyl Pyridine)/Poly (Vinyl Phenol) Revisited

**DOI:** 10.3390/polym11071153

**Published:** 2019-07-05

**Authors:** Osamu Urakawa, Ayaka Yasue

**Affiliations:** Department of Macromolecular Science, Graduate School of Science, Osaka University, 1-1 Machikaneyama, Toyonaka, Osaka 560-0043, Japan

**Keywords:** hydrogen bond, miscible blend, glass transition temperature, poly (vinyl phenol), poly (vinyl pyridine), Kwei equation

## Abstract

We examined the composition and molecular weight dependence of the glass transition temperature in detail for two types of hydrogen bonding miscible blends: poly (2-vinyl pyridine)/poly (vinyl phenol) (2VPy/VPh) and poly (4-vinyl pyridine)/poly (vinyl phenol) (4VPy/VPh). Regarding the functional form of the glass transition temperature, *T*_g_, as a function of the weight fraction, we found a weak deviation from the Kwei equation for 2VPy/VPh blends. In contrast, such a deviation was not observed for the 4VPy/VPh blend. By relating the difference in the functional forms of *T*_g_ between the two blend systems to the difference in hydrogen bonding ability, we proposed a modified version of the Kwei equation. As for the interaction parameter, *q* in the Kwei equation, clear molecular weight dependence was observed for 2VPy/VPh blends: the lower the VPh molecular weight in the oligomer level, the higher the *q* values, suggesting the higher hydrogen bonding formability near the polymer chain ends than the middle part of a polymer chain.

## 1. Introduction

Polymer alloy is a widely used concept and still irreplaceable method to control the physical properties of polymer materials [1,2]. Several alloys examined so far are roughly divided into two types: phase-separated blends and miscible blends. In the former case, not only the properties of each phase but also the morphology and the interfacial energy often play an important role in determining their properties; [3,4] in the latter case the interaction between two components becomes important in addition to the intrinsic properties of the component polymers [5,6,7]. Especially, for miscible blends with strong attractive interaction, such as hydrogen bonds (H-bonds), the H-bond formation greatly modifies their properties compared to the blend without specific interaction [8,9,10,11]. Many studies on thermal and mechanical properties of H-bonding polymer blends have been reported so far to relate their properties to the H-bonding interaction strength [12,13,14]. Specifically, glass transition temperatures, *T*_g_, of such blends become higher in some cases than the pure component *T*_g_ due to the attractive interaction between the component polymers [15,16,17,18,19,20,21]. On top of that, studies about a new class of materials by combining the micro-phase separation and the H-bonding interaction have been reported [22,23,24,25,26,27].

To represent the composition dependence of *T*_g_ for polymer blends, polymer 1/polymer 2, with specific interaction, Kwei proposed an empirical equation which involves the interaction term [16].
(1)$$T_{g}=\frac{w_{1}\;T_{g1}+kw_{2}T_{g2}}{w_{1}+kw_{2}}+qw_{1}w_{2}$$

Here, $w_{1}$ and $w_{2}\;\left(=1-w_{1}\right)$ are the weight fractions of the component polymers, *k* and *q* are parameters. This equation has been widely used to describe the composition dependence of *T*_g_ not only for polymer blends but also for copolymers [15,16,17,18,19,20,21,28,29]. The parameter *q* in the second term corresponds to the strength of the inter-polymer interaction: stronger attractive interaction is represented by the larger *q* value. This parameter is often utilized to quantitatively evaluate the interaction strength between component polymers especially for H-bonding polymer blends. The first term in Equation (1) is the same with the Gordon-Taylor (GT) expression [30]. For the physical meaning of the parameter *k*, two interpretations have been suggested: One interpretation is to regard it as the ratio of the volume expansion coefficients of the pure components at *T*_g_ [30,31], and the other is to regard it as the ratio of the heat capacity increments of the pure components [31,32]. Regardless of such physical meanings of *k*, this parameter has been often used as just an adjustable parameter and has sometimes taken unphysically large or small values (k>10 or k<0.1) especially to fit the S-shaped $T_{g}\left(w_{1}\right)$ curves having inflection point [16,33]. In other words, because the interaction term, $qw_{1}\left(1-w_{1}\right)$, in Equation (1) has symmetric shape as a function of the composition, it is necessary to change the *k* value significantly for the reproduction of $T_{g}\left(w_{1}\right)$ curves when its shape is asymmetrically skewed. As is well known, the H-bond formation is a dynamical process and the reaction between proton donor and acceptor sometimes takes a long time to reach equilibrium, especially for solid polymer materials [34]. Therefore, it will always be necessary to check whether the experimental value is at equilibrium or not. Furthermore, since some H-bonding species, such as a hydroxyl group, can self-associate [9,10,35], the competing effect of the self-association versus hetero-association leads to the change in the effective concentrations of the proton donors and the acceptors. Such effective values should replace the actual composition in Equation (1), leading to the change of the interaction term.

In this paper, we focus on the functional forms of $T_{g}\left(w_{1}\right)$ in detail, and how they depend on the H-bonding strength. We have examined two blend systems: poly (2-vinyl pyridine)/poly (vinyl phenol) (2VPy/VPh) and poly (4-vinyl pyridine)/VPh (4VPy/VPh) in which hydrogen bonds are formed as schematically shown in Figure 1. In addition to these structures, VPh component can self-associate through the OH···OH hydrogen bond. Further noted is that the ability as proton acceptor for 4VPy is known to be higher than 2VPy [18,36]. For these systems, the critical test of Equation (1) is conducted especially including the case of $T_{g1}=T_{g2}$ in which case the parameter *k* in Equation (1) is no longer an adjustable parameter because of the cancelation of the numerator and the denominator of the first term. We found that Equation (1) could fit the $T_{g}\left(w_{1}\right)$ data of 4VPy/VPh but could not fit those of 2VPy/VPh blends. The reason for these phenomena is considered in connection with the H-bonding structure and the modification of Equation (1) is proposed.

## 2. Materials and Methods

Poly (vinyl phenol) samples with molecular weights of 1.9 × 10^3^ and 4.5 × 10^3^, denoted as VPh2 and VPh5, were supplied by JSR corporation. Poly (vinyl phenol) with molecular weight of 1.6 × 10^4^, denoted as VPh16, was supplied by A. Takano and Y. Mastushita in Nagoya University [37]. Poly (2-vinyl pyridine) was purchased from Scientific Polymer Products, Inc. Poly (4-vinyl pyridine) was purchased from Sigma-Aldrich. All the sample characteristics are summarized in Table 1. 

The blend samples were prepared by the following two methods. Method (I): Dioxane solutions of VPy and VPh were separately prepared with various concentrations from 0.4 to 1.3 wt%, and then mixed together. We adjusted the total polymer concentration in the mixed solution to be 1.0 ± 0.1 wt%. Just after mixing two solutions, solid polymer mixture was precipitated because of the formation of H-bonded complex in dioxane. Method (II): Prescribed amounts of VPy and VPh samples were dissolved in dimethyl sulfoxide (DMSO), which can dissociate interpolymer H-bonds, with total polymer concentrations to be around 10 wt%, and then the obtained homogeneous ternary solutions were re-precipitated in deionized water. The precipitates obtained by the above-mentioned two methods were recovered by centrifugation and washed by dioxane or water and then dried in vacuum for one day at 120 °C. The precise blend compositions were determined by the elemental analysis method with organic elemental analyzer (MT-6, YANACO, Tokyo, Japan) which simultaneously determined the amounts of C, H, N elements. In this study we mainly used the Method (II) to prepare the blend samples. Only for VPy40/VPh16 samples both methods were used to see the difference.

The glass transition temperature, *T*_g_ and the heat capacity increment, *C*_p_ at *T*_g_ for all the homopolymers and blend samples were determined with a differential scanning calorimeter (DSC 6220, Seiko Instruments Inc., Chiba, Japan). The rate of both heating and cooling processes was 10 °C min^−1^. Three thermal scan cycles of 1st heating–1st cooling–2nd heating–2nd cooling–3rd heating were conducted in the temperature range of 30 °C–230 °C. The *T*_g_ value was determined in the 2nd heating scan as the temperature at which the time derivative of the heat flow curve exhibited a peak corresponding the inflection point (nearly equal to the midpoint) of the heat flow jump. In the 3rd heating, we checked all the data to agree with that of the 2nd heating scan. The *T*_g_ and the *C*_p_ data for homopolymers are also shown in Table 1. 

## 3. Results and Discussion

### 3.1. Weight Fractions in 2VPy/VPh Blend Obtained by Two Preparation Methods

The weight fractions of 2VPy40, *w*_2VPy_, in the blend of 2VPy40/VPh16 obtained by the Methods (I) and (II) are shown in Figure 2 as functions of the weight fractions in the initial feed solutions before precipitation. It is seen that the *w*_2VPy_ in precipitates prepared by Method (I) are in the narrow composition range from 0.4 to 0.75 in spite of the large variation of the composition in the feed solutions. This suggests that the 2VPy and VPh prefer to form 1:1 complexes in dioxane solutions. The amounts of precipitates, i.e., conversions, were also the highest for the 1:1 feed solutions, supporting the idea that 1:1 complex formation is favorable [38]. These results are in harmony with the data reported by Dai et al. who used ethanol solutions of 2VPy and VPh to obtain the precipitates [18]. As shown in Figure 2, for the samples prepared by Method (II), on the other hand, the compositions in precipitates are almost the same with those in the feed solutions. 

For the preparation of other blend system; 2VPy40/VPh2, 2VPy40/VPh5, and 4VPy60/VPh5, we only use the Method (II). The compositions determined by elemental analysis exhibited similar trend with that shown in Figure 2: The determined weight fractions were almost exactly the same with those of the feed solutions. Hereafter, we used the values determined by elemental analysis as the blend compositions, *w*_2VPy_ or *w*_4VPy_. 

### 3.2. Glass Transition Behavior

For all the blends with different compositions, we observed single glass transition by DSC. This indicates that all the blends are homogeneously mixed at the molecular level. As an example, the DSC traces of 2VPy40/VPh16 blends prepared by Methods (I) and (II) are shown in Figure 3. The transitions are single but the transition widths are dependent on the composition, probably reflecting different degree of the composition heterogeneity. It is noted that the transition widths are comparable with those of miscible blends without specific interaction [39,40]. The detailed data of the transition width including other blend systems are shown in the supporting information (Appendix A).

Regarding the shapes of the transitions, it is seen that some DSC data, especially for the blend samples with high *T*_g_ values, exhibit an endothermic peak due to the enthalpy relaxation. This may be because the high *T*_g_ samples experience longer aging times below their *T*_g_ during the temperature scan cycles of DSC measurements. Different degrees of aging of the blend samples make the comparison of *T*_g_ values difficult, since longer aging time generally leads to the increase in the *T*_g_ values. To check such effect on the *T*_g_ values, we analyzed the cooling scan DSC data, which is not affected by the physical aging. Appendix Aa–d (in the supporting information) compare the heating and cooling *T*_g_ values for all the blend systems. It can be seen that the *T*_g_ values are slightly different between heating and cooling scans, but the functional forms of *T*_g_ (*w*_1_) are almost the same. Namely, the effect of aging on the *T*_g_ values is limited and does not disturb the argument being made in the next section about the composition dependence of *T*_g_. 

### 3.3. Composition Dependence of T_g_

Figure 4 compares the composition dependence of *T*_g_ for 2VPy40/VPh16 blend prepared by two different methods, Methods (I) and (II). No systematic difference in *T*_g_ for two series of samples is seen, meaning that both samples have similar H-bonding structures in spite of the different preparation methods. Dai et al. [18] reported different results for 2VPy/VPh, 4VPy/VPh, and 4VPy-styrene copolymer/VPy blends: higher *T*_g_ for the samples precipitated by mixing ethanol solution (similar to the Method (I)) than for the samples cast from the dimethylformamide (DMF) ternary solutions (similar to the Method (II)). Although the solvents they used are different, at present the reason for the different results is unknown. However, our data suggest that the two preparation methods used in this study yield samples with almost the same hydrogen bonding structures.

The *w*_2VPy_ dependence of *T*_g_ for 2VPy40/VPh16 was fitted with the Kwei equation (Equation (1)) in our previous study [38]. In Figure 3, the fitting results is again shown. In this case, *k* = 1.28 was calculated following the Wood’s formula: $k=\Delta C_{p2}/\Delta C_{p1}$, [31,32] and the *q* was used as only the floating parameter, resulting in *q* = 120 °C. In this figure, the average *T*_g_ calculated by the following Gordon Taylor equation [30] is also shown.
(2)$$T_{g}=\frac{w_{1}\;T_{g1}+kw_{2}T_{g2}}{w_{1}+kw_{2}}$$

The subscript numbers, 1 and 2 in Equations (1) and (2) correspond to 2VPy and VPh, respectively. As is seen in Figure 3, Kwei equation can fit the data fairly well. However, it is seen that the calculated curve slightly deviates from the data points in the low and high *w*_2VPy_ region. This small discrepancy could not be reduced any further even by adjusting the two parameters: *q* and *k*. This trend is also apparent for the different molecular weight (MW) samples as will be shown later. It is noted that the expression of Equations (1) and (2) use the weight fraction instead of volume fraction, but which fraction to use may be debatable. However, because the densities of VPy and VPh are nearly equal (1.16 g cm^−3^ and 1.15 g cm^−3^, respectively), it is not necessary for these blends to distinguish between volume and weight fractions. 

Figure 5 shows the $T_{g}\left(w_{2VPy}\right)$ data along with the fit-results by Equation (1) for the blends of (a) 2VPy40/VPh5 and (b) 2VPy40/VPh2, which were prepared by the Method (II). Here, the *k* calculated as $\Delta C_{p,\mathrm{VPh}}/\Delta C_{p,\mathrm{VPy}}$ and *q* values used as parameters to fit the data were as follows: (a) *k* = 1.28, *q* = 158 °C and (b) *k* = 1.35 and *q* = 186 °C. In these figures, a similar type of discrepancy as seen in Figure 4 can be recognized between the data and the Kwei equation. Especially, for the 2VPy40/VPh2 blend, the pure component *T*_g_s are almost the same, so that the parameter *k* disappears from Equation (1) and *q* becomes the only parameter. Therefore, this can be a critical test for the functional form of $T_{g}\left(w_{2VPy}\right)$, and indicates the apparent failure of Equation (1). 

All the data shown in Figure 4 and Figure 5 clearly indicate the systematic deviation of the Kwei equation from the data for 2VPy/VPh blends: The experimental $T_{g}\left(w_{2VPy}\right)$ data are skewed to the low $w_{2VPy}$ side compared with the fit curves. By considering that 2VPy and VPh forms 1:1 stoichiometric H-bond, the *T*_g_ enhancement represented by the second term of Equation (1) should take a maximum at $w_{2\mathrm{VPy}}\sim 0.5$. One possible reason for the deviation toward the low $w_{2\mathrm{VPy}}$ side is that the effective weight fraction of VPh is lower than the actual one because of the self-association of VPh component. Figure 6 schematically shows the image of the hydrogen bonding structures formed in this system. Since hydroxyl OH groups can form self-associated OH···OH dimer, and OH···OH···OH··· multimers [9,10,35,41], the effective concentration of the OH groups which can be used to form hetero-type H-bonds, OH···N, will decrease. The quantitative description of the fractions of these H-bonding spices may be possible by using association constants for these self-associates and hetero-associates [35,41]. However, the way to incorporate the fraction of such H-bonding spices including their temperature dependence to the Kwei equation has not been clarified because this equation is just an empirical formula. 

To consider the above-mentioned competitive association, here we simply introduce a new parameter, into Equation (1) just phenomenologically. We assume that the effective weight fraction of component 2 (VPh) is represented by $w_{2}^{\alpha }$ (when α>1, $w_{2}^{\alpha }<w_{2}$) and modify the Kwei equation as follows:(3)$$T_{g}=\frac{w_{1}\;T_{g1}+kw_{2}T_{g2}}{w_{1}+kw_{2}}+qw_{1}^{2-\alpha }w_{2}^{\alpha }$$

In this equation, the exponent of $w_{1}$ is assumed to be 2−α so that the interaction parameter, *q*, has the similar value as the original Kwei equation. The fitted result with Equation (3) are shown in Figure 4 and Figure 5 by dashed lines. It is seen that Equation (3) better fits the experimental data than Equation (1). Table 2 summarizes the α and *q* values used in this fitting. It is seen that the *q* values were almost the same as those determined by the fitting with the original Kwei equation. The α value seems to be independent of the molecular weight of VPh when compared among the three 2VPy40/VPh blends. On the other hand, clear molecular weight (MW) dependence is seen for the interaction parameter, *q*: By lowering the MW of VPh, the *q* value becomes larger. To the best of our knowledge, MW dependence of *q* has not been fully examined so far. This observed trend may be due to the difference in hydrogen bonding ability between the polymer chain ends and the center part. It is well known that the decrease in *T*_g_ in the oligomer region is due to the increase of the concentration of the chain ends which have higher mobility and lower *T*_g_ than the middle part [42,43,44]. If the chain ends form intermolecular hydrogen bond more effectively than the middle part because of the less steric hindrance, the motion of the mobile chain ends is largely suppressed, resulting in the larger increase of *T*_g_ for low MW blends compared with their non-H-bonded state. This is likely to be the cause of the increase in *q* values in the low molecular weight region.

In order to investigate the intrinsic difference of the H-bonding ability, we examined the composition dependence of *T*_g_ for the 4VPy60/VPh5 blend, since the H-bonding strength of 4VPy is higher than 2VPy due to the less steric hindrance of the pyridine ring [18]. Figure 7 shows the $T_{g}\left(w_{4\mathrm{VPy}}\right)$ data and the fit-results with Equations (1) and (3). The fitting parameters have already shown in Table 2. For this blend, the pure component *T*_g_s are almost the same, and thus the parameter *k* does not affect the shapes of the fitting curves. In this critical test, it is found that the $T_{g}\left(w_{4\mathrm{VPy}}\right)$ has nearly a symmetric shape, so that the *α* value is nearly equal to unity: α=1.02. This result possibly means that the effective compositions are nearly the same with the actual compositions due to the more favorable formation of the hetero-association. Concerning the *q* value of this system (*q* = 175 °C), it is higher than that of VPh5/2VPy40 (*q* = 154 °C), which has comparable MW combination. This trend is consistent with the data reported previously [18].

Both the *q* and *α* are related to the hetero-association formability and thus it is naturally thought that these parameters may be correlated. However, by comparing the *q* and *α* values shown in Table 2 including both VPh/2VPy and VPh/4VPy systems, the larger *q* does not always result in the smaller *α*. This is because two factors, MW and chemical structure difference, are involved in this comparison. The α values differ between 2VPy/VPh and 4VPy/VPh but they are nearly independent of MW. In contrast, the *q* values are dependent on both factors. As discussed earlier, the α parameter reflects the effective concentration of the proton donors (OH) which is modified by the effect of self-association. In other words, the *α* will be determined as a result of the competition of the self-association and hetero-association: α=1 means negligible contribution of the self-association of the OH component but α>1 means that the self-association effectively works to reduce the number of hetero-association. In contrast, the *q* value reflects only the hetero-association formability. Therefore, we think that the evaluation of both two parameters, *α* and *q*, will give important information to consider the H-bonding structure in more detail.

## 4. Conclusions

Glass transition temperatures as functions of the blend composition and the molecular weight have been examined in detail for two kinds of hydrogen bonding miscible blends: 2VPy/VPh and 4VPy/VPh. The Kwei’s empirical equation could not reproduce the functional forms of *T*_g_ (*w*_2VPy_) for 2VPy/VPh blends in the low and high *w*_2VPy_ regions, but worked well for a 4VPy/VPh blend. By considering the difference in the degree of the self-association of the VPh component between the two systems, the idea of effective composition, which can actually contribute to the H-bonding between VPy–VPh, has been introduced to modify the Kwei equation with keeping the interaction parameter, *q*, to be quantitatively the same. The proposed empirical equation could well reproduce the functional forms of *T*_g_ (*w*) for all the blend systems. 

Regarding the interaction parameter, *q*, we have found clear molecular weight dependence in 2VPy/VPh blends: The lower the VPh molecular weight, the higher the *q* values. From this result, one possibility has been proposed: H-bonding formation is more effective at polymer chain ends than the center part, and thus the lower MW blends, which have larger chain ends contribution, have higher *q*.

## Figures and Tables

**Figure 1 polymers-11-01153-f001:**
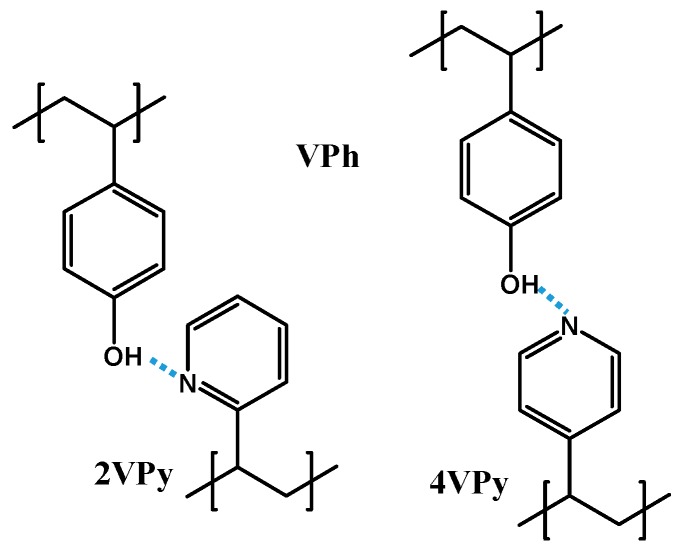
Hydrogen bonding structures of 2VPy/VPh and 4VPy/VPh.

**Figure 2 polymers-11-01153-f002:**
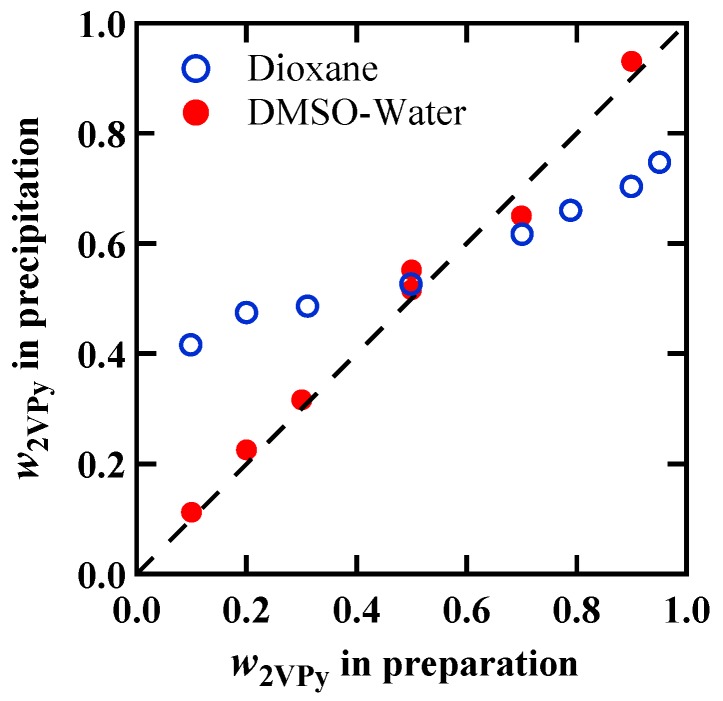
Weight fractions of 2VPy40 in finally obtained 2VPy40/VPh16 blends determined by the elemental analysis method plotted against the weight fractions determined by the weighing in the solutions preparation. The data of two kinds of blend samples prepared by Method (I) (precipitated by mixing two dioxane solutions) and Method (II) (precipitated in water from the DMSO ternary solutions) are shown.

**Figure 3 polymers-11-01153-f003:**
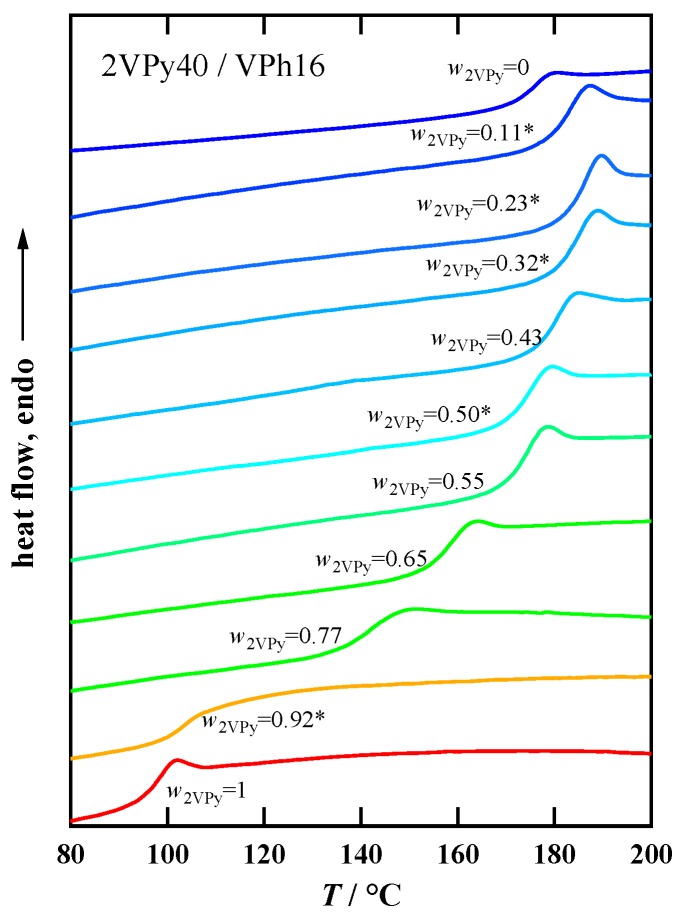
Examples of the 2nd DSC heating traces for 2VPy40/VPh16 blends with several compositions prepared by Methods (I) and (II). The asterisk symbols indicate the samples prepared by Method (II).

**Figure 4 polymers-11-01153-f004:**
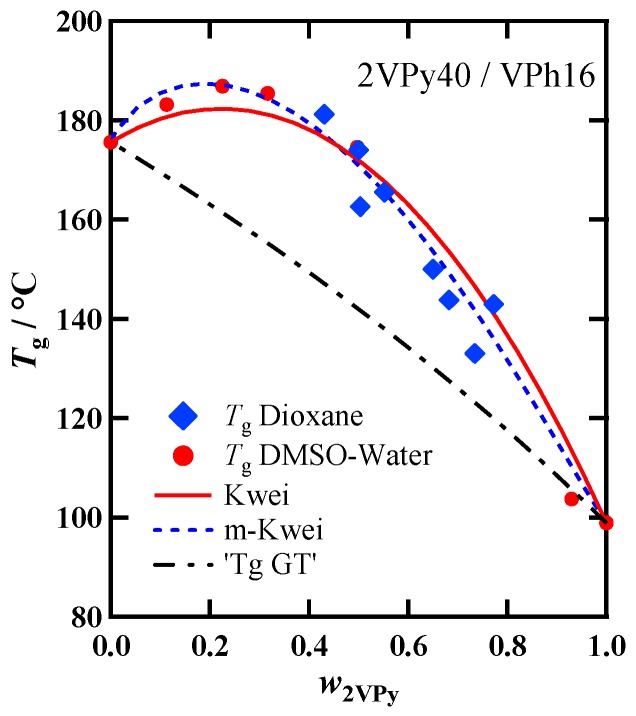
Dependence of *T*_g_ on the weight fraction of 2VPy for 2VPy40/VPh16 blends prepared by Method (I) and (II). The fit-results with Equation (1) (denoted as Kwei) and Equation (3) (denoted as m-Kwei) are shown by solid and dashed lines, respectively. The calculated result with Equation (2) (denoted as GT) is also shown by dash-dotted line.

**Figure 5 polymers-11-01153-f005:**
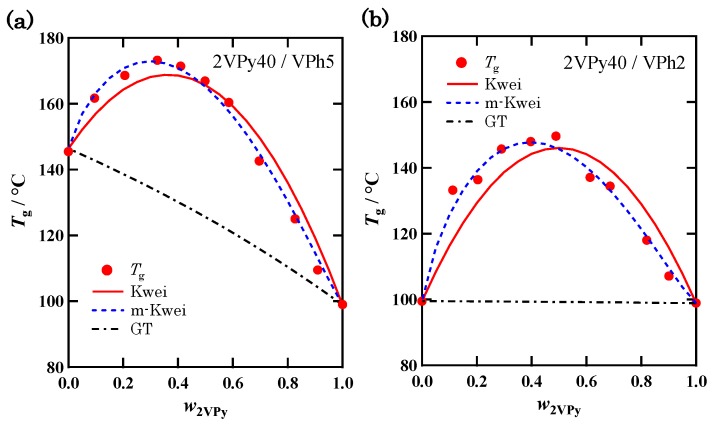
Dependence of *T*_g_ on the weight fraction of 2VPy for (**a**) 2VPy40/VPh5 and (**b**) 2VPy40/VPh2 blends prepared by Method (II). The fitted (or calculated) results with Equations (1)–(3) are shown by solid, dash-dotted, and dashed lines, respectively.

**Figure 6 polymers-11-01153-f006:**
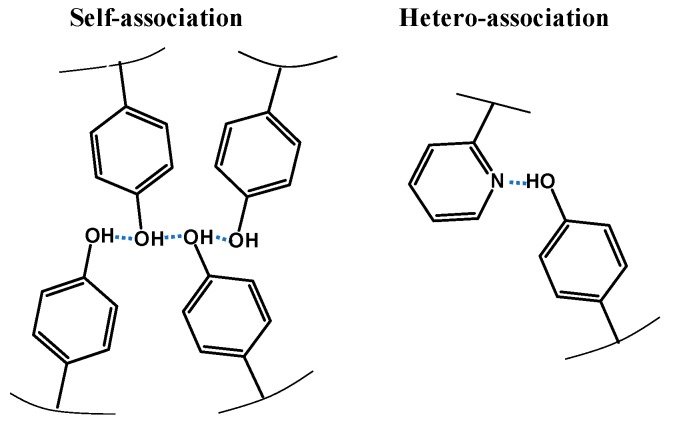
Schematic illustration of possible hydrogen bonding structure of self-association and hetero-association.

**Figure 7 polymers-11-01153-f007:**
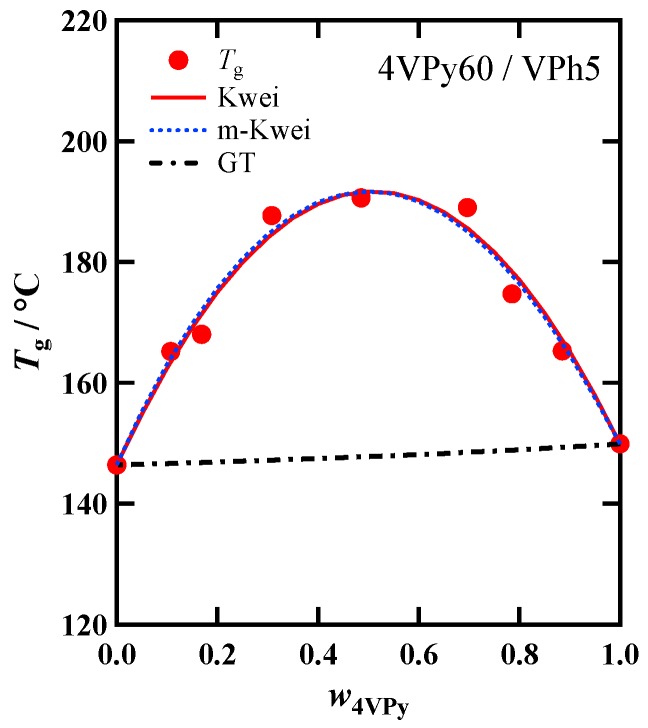
Dependence of *T*_g_ on the weight fraction of 4VPy for 4VPy60/VPh5 blend prepared by Method (II). The fitted (or calculated) results with Equations (1)–(3) are shown by solid, dash-dotted, and dashed lines, respectively.

**Table 1 polymers-11-01153-t001:** Characteristics of poly (vinyl phenol) (VPh), poly (2-vinyl pyridine) (2VPy), and poly (4-vinyl pyridine) (4VPy) samples.

Code	10^3^ *M*_w_	*M*_w_/*M*_n_	*T*_g_/°C	*C*_p_/J °C^−1^ g^−1^
VPh2	1.9		99.8	0.47
VPh5	4.5	1.8	146	0.44
VPh16 ^(a)^	16	1.8	177	0.44
2VPy40	39 ^(b)^	1.03	99.0	0.34
4VPy60	60 ^(c)^		148	0.29

^(a)^ This is the similar series of samples used in reference [37]. We previously reported a wrong number for the molecular weight [38]. ^(b)^ Determined from the intrinsic viscosity (viscosity averaged molecular weight). ^(c)^ Manufacture’s value.

**Table 2 polymers-11-01153-t002:** Parameters of Equation (3) used to reproduce the weight fraction dependence of *T*_g_ for the four blend systems.

Sample	$k=\Delta C_{p2}/\Delta C_{p1}$	*q* (Equation (1))/°C	*q* (Equation (3))/°C	*α* (Equation (3))
2VPy40/VPh16	1.28 ± 0.1	120 ± 1	116 ± 1	1.19 ± 0.02
2VPy40/VPh5	1.28 ± 0.1	160 ± 1	158 ± 1	1.17 ± 0.02
2VPy40/VPh2	1.35 ± 0.1	187 ± 1	186 ± 1	1.18 ± 0.02
n4VPy60/VPh5	1.52 ± 0.1	176 ± 1	176 ± 1	1.02 ± 0.02

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
