# Peer review of "Glass Transition Behaviors of Poly (Vinyl Pyridine)/Poly (Vinyl Phenol) Revisited"

_polymers, 2019, doi:10.3390/polym11071153_

Reviewer 1 Report

This is a well written paper (outside of some basic English usage errors expected of non-native speakers).  The primary experimental finding is that the Tg of the mixtures is higher for intermediate mix ratios than either of the extremes.  The results are then fit to the Kwei equation.  The experiments appear to be carefully performed, however I do have some reservations about the DSC characterization.

I would like the authors to consider the following points.

1) In figure 3, the Tg’s for w2 = 0.11, 0.23, and 0.31 are all higher than the terminal w2=0 which makes a simple mixing rule unable to fit the results (requiring the qw1w2 term).  My concern is that the peaks associated with the transition are more pronounced than for the w2=0 case.  Since a peak indicates physical aging, this implies that these blends are more highly physically aged than w2=0.  Another aspect of physical aging is that the location of the step is moved to higher temperature (i.e., the apparent mid-point Tg is increased).  My first conclusion from looking at Fig 3 is that there is more dramatic physical aging in the w2 = 0.11, 0.23, and 0.31 than the w2=0 case and that I couldn’t draw conclusions about Tg (because of the aging).  That is that the mid-point Tg deduced from the heating curves is not the “true” Tg since it is perturbed by physical aging.

2) I suggest that the authors take Tg from the cooling curves in addition to (or instead of) the ones from the heating curves.  These values will be lower than the heating Tg’s, but will not be perturbed by physical aging.  Are the Tg’s(cooling) still higher for w2 = 0.11, 0.23, and 0.31 than for w2=0?  

3) Add to the figure caption for fig 3 that these are for the second heating ramp (or are they for the 3rd ramp?)

4) Question on the Kwei equation.  The qw1w2 is similar in spirit to the interaction term in the Flory-Huggins equation.  However, in the Flory Huggins equation, volume fractions are used instead of weight fractions.  Are the weight fractions the same as the volume fractions for these materials? If not, I would recommend trying volume fractions to see if that improves the fit.  To this end, adding a column for density to table 1 would be useful.

5) The authors’ are extracting Tg’s from DSC curves.  It would be useful to be able to see additional thermograms – If this journal permits supplementary materials, I would suggest that most of the DSC curves be archived there (heating and cooling for each of the cases).

Author Response

Thank you for carefully reading our manuscript and giving valuable comments.

1) In figure 3, the Tg’s for w2 = 0.11, 0.23, and 0.31 are all higher than the terminal w2=0 which makes a simple mixing rule unable to fit the results (requiring the qw1w2 term).  My concern is that the peaks associated with the transition are more pronounced than for the w2=0 case.  Since a peak indicates physical aging, this implies that these blends are more highly physically aged than w2=0.  Another aspect of physical aging is that the location of the step is moved to higher temperature (i.e., the apparent mid-point Tg is increased).  My first conclusion from looking at Fig 3 is that there is more dramatic physical aging in the w2 = 0.11, 0.23, and 0.31 than the w2=0 case and that I couldn’t draw conclusions about Tg (because of the aging).  That is that the mid-point Tg deduced from the heating curves is not the “true” Tg since it is perturbed by physical aging.

2) I suggest that the authors take Tg from the cooling curves in addition to (or instead of) the ones from the heating curves.  These values will be lower than the heating Tg’s, but will not be perturbed by physical aging.  Are the Tg’s(cooling) still higher for w2 = 0.11, 0.23, and 0.31 than for w2=0?  

We understand the reviewer’s concern that the degrees of aging differ sample to sample, which can affect the Tg values and thus the functional form of Tg(w). Following the reviewer’s suggestion (2), the effect of aging has been examined by using the cooling Tg data. The comparison between the heating and cooling Tgs have been added in the supporting information (Figure S3). Figure S3 indicates that the effect of aging is not so important as to significantly change the functional form of Tg(w). This explanation has been added on page 4 (line 131-140).

As another problem, we found that the cooling DSC curves are broader than the heating curves (cf. Figure S1), so that the method of Tg determination from the mid-point of the transition curve causes uncertainty. Therefore, we have changed the method of determining Tg from the peak of the DDSC (time derivative of the DSC curve). This explanation has been added in the experimental section.

3) Add to the figure caption for fig 3 that these are for the second heating ramp (or are they for the 3rd ramp?)

Figure caption has been revised.

4) Question on the Kwei equation.  The qw1w2 is similar in spirit to the interaction term in the Flory-Huggins equation.  However, in the Flory Huggins equation, volume fractions are used instead of weight fractions.  Are the weight fractions the same as the volume fractions for these materials? If not, I would recommend trying volume fractions to see if that improves the fit.  To this end, adding a column for density to table 1 would be useful.

The Kwei equation is originally written in terms of weight fraction, so that we followed the original expression. Fortunately, the densities of poly(vinyl pyridine) and poly(vinyl phenol) are almost the same: 1.16 g/ml (VPy) and 1.15 g/ml (VPh). Therefore, we don’t need to distinguish between the weight fraction and the volume fraction. This comment is added in the manuscript (page 6, line 168-171).   

5) The authors’ are extracting Tg’s from DSC curves.  It would be useful to be able to see additional thermograms – If this journal permits supplementary materials, I would suggest that most of the DSC curves be archived there (heating and cooling for each of the cases).

Most of the thermograms have been added in the supporting information (Figure S2).

Reviewer 2 Report

The manuscript reports calorimetric data for miscible blends of 2 and 4 poly(vinyl pyridine) with poly(vinyl phenol). In doing so, the authors assess the concentration dependence of the glass transition temperature (Tg) and find that the well-known Kwei equation is unable to provide a completely satisfactory quantitative fitting of these Tg data. To overcome such inability, the authors propose to introduce an additional term to the Kwei equation based on a parameter “q”, which accounts for the self-association of each component of the blend.

Although the approach employed in the present work is completely phenomenological and introducing new fitting parameters always implies more accurate fits, the outcome of such fits appears to be reasonable and physically sound. Apart from the mere data fitting, I suggest that the authors make an effort to propose a framework to make the new proposed equation predictive. For instance, what would it happen if one of the components of the blend of the present study was mixed with another polymer? Could any Tg prediction be done using the q parameter obtained in the model mixture of the present study?

Apart from the previous point, the following minor point needs to be addressed. The glass transition encompasses a wide temperature range. This is particularly crucial in blends. For instance for the blend with w_2VPy = 0.77 a glass transition range of about 20 K can be inferred. Rather than a single point, I suggest to introduce a range in Figures 4, 5 and 7.

Author Response

Thank you for a very suggestive comments.

The manuscript reports calorimetric data for miscible blends of 2 and 4 poly(vinyl pyridine) with poly(vinyl phenol). In doing so, the authors assess the concentration dependence of the glass transition temperature (Tg) and find that the well-known Kwei equation is unable to provide a completely satisfactory quantitative fitting of these Tg data. To overcome such inability, the authors propose to introduce an additional term to the Kwei equation based on a parameter “q”, which accounts for the self-association of each component of the blend.

Although the approach employed in the present work is completely phenomenological and introducing new fitting parameters always implies more accurate fits, the outcome of such fits appears to be reasonable and physically sound. Apart from the mere data fitting, I suggest that the authors make an effort to propose a framework to make the new proposed equation predictive. For instance, what would it happen if one of the components of the blend of the present study was mixed with another polymer? Could any Tg prediction be done using the q parameter obtained in the model mixture of the present study?

The parameter a introduced in eq.(3) reflects the effective concentration of the proton donors (OH) which is modified by the effect of self-association due to its self-association ability. This parameter is also affected by the nature of the hydrogen bonding partner (pyridine). In short, the a parameter is determined by both the self- and hetero-association ability. In contrast, the parameter q is determined by the ability of the hetero-association formation. Therefore, these two parameters are not intrinsic values to a single component polymer. Consequently, it is difficult to predict the Tg values when the blend partner is changed even if the a and q values for each component with different partner are known. In the revised manuscript, the physical meaning of the new parameter a and the importance to determine this parameter is described in more detail on page 8 line 237-245.

Apart from the previous point, the following minor point needs to be addressed. The glass transition encompasses a wide temperature range. This is particularly crucial in blends. For instance for the blend with w_2VPy = 0.77 a glass transition range of about 20 K can be inferred. Rather than a single point, I suggest to introduce a range in Figures 4, 5 and 7.

In the supporting information (Figure S3), the widths of the transition for all the samples have been added.

Reviewer 3 Report

Authors present results from experiments on miscible blends characterized by hydrogen bonding. They place focus on examining the effect of molecular weight and composition (fraction) on the glass transition temperature. Prominent among the result is that the aforementioned effect can be captured accurately by introducing a modified version of the Kwei formula instead of the original version which shows deviations. Explanation of observed trends is based on the formation of hydrogen bonds near the ends versus the inner chain segments.

The method is well described and the results are adequately explained. The topic is of general interest and is expected to draw attention from the readership of Polymers. Subject to minor revisions the manuscript would be thus suitable for publication.

Below is the list of my comments and questions.

) It is not fully clear if in all reported cases q comes out as an adjustable parameter of the fitting formula or not (for example the value of 114 oC in line 139 versus the statement in line 146). With respect to k is it calculated always from the Wood´s expression?

) According to the data of Table 2 it seems that monotonically increased q values are not accompanied by  corresponding reduction of the exponent alpha values. However, based on the physical interpretation given by the authors on the alpha parameter this is not immediately apparent. The authors should elaborate on this point as it is central to the core of the present manuscript. Also, error bars could be added in the reported values of Table 2.

) Correspondence of symbols should be mentioned when first introduced even if their meaning is trivial (for example T_g in the abstract).

) Meaning of sentence in line 50 is unclear and the phrase should be restated ("it is necessary largely to vary").

) To be consistent with the legend of Fig. 2, in Fig. 3 "diox precipitated" should be changed to "Dioxane".

) Manuscript has some syntax and grammar errors that need to be corrected. For example line 24: "is roughly" -> "are roughly". Line 47: "was sometimes" -> "has sometimes".

) Article and journal titles are incorrect in Ref. 30.

Author Response

Thank you for carefully reading our manuscript. We appreciate your valuable comments.

) It is not fully clear if in all reported cases q comes out as an adjustable parameter of the fitting formula or not (for example the value of 114 oC in line 139 versus the statement in line 146). With respect to k is it calculated always from the Wood´s expression?

All the cases q value came out as an adjustable parameter, and k is calculated always form the Wood’s expression. To clarify the fitting procedure, the corresponding parts (line 160-162, Table 2) were revised.

) According to the data of Table 2 it seems that monotonically increased q values are not accompanied by corresponding reduction of the exponent alpha values. However, based on the physical interpretation given by the authors on the alpha parameter this is not immediately apparent. The authors should elaborate on this point as it is central to the core of the present manuscript. Also, error bars could be added in the reported values of Table 2.

We have changed the method of determining Tg: Tg was determined from the peak of the DDSC (time derivative of the DSC curve) in order to eliminate the ambiguity due to the base line fluctuation, etc. Because of this change the fitting results, parameters a and q values, have slightly changed. Table 2 shows that the a values are almost independent of the molecular weight. We think that a and q are not necessarily correlated because these two parameters have slightly different origins: the a parameter reflects the effective concentration of the proton donners (OH) which is modified by the effect of self-association. In other word, the a will be determined as a result of the competition of the self-association and hetero-association: α = 1 means negligible contribution of the self-association of the OH component but a > 1 means that the self-association effectively works to reduce the number of hetero-association. In contrast, the q value reflects only the hetero-association formability. This explanation has been added on page 8, line 238-245. Error bars due to the fitting have been added in the values of Table 2.

) Correspondence of symbols should be mentioned when first introduced even if their meaning is trivial (for example Tg in the abstract).

 The indicated part of the abstract has been revised.

) Meaning of sentence in line 50 is unclear and the phrase should be restated ("it is necessary largely to vary").

 The sentence in line 52 of the new manuscript have been corrected. (“it is necessary to change the k value significantly”).

) To be consistent with the legend of Fig. 2, in Fig. 3 "diox precipitated" should be changed to "Dioxane".

 We corrected the text in Figure 3.  

) Manuscript has some syntax and grammar errors that need to be corrected. For example line 24: "is roughly" -> "are roughly". Line 47: "was sometimes" -> "has sometimes".

 We have corrected the syntax and grammar errors as much as possible.

) Article and journal titles are incorrect in Ref. 30.

We think the article and the journal title of Ref.30 (Ref.38 in the new manuscript) is correct. Because this article is written in Japanese, it may be difficult to access the web site. We added the DOI information of this reference to facilitate the access.

The URL is as follows.

 https://www.jstage.jst.go.jp/article/jsms/64/1/64_43/_article